# OpenReview forum: "Evaluating from Benign to Dynamic Adversarial: A Squid Game for Large Language Models"
_ICLR.cc/2026/Conference — ICLR 2026 Conference Withdrawn Submission_

### Official Review · Reviewer_p4bi · 2025-10-30

**Soundness:** 2
**Presentation:** 3
**Contribution:** 1
**Rating:** 2
**Confidence:** 4

**Summary:**

The work is a benchmarking framework to evaluate LLMs through multi-agent competitive interactions. It proposes an arena structure where individual LLMs are cast as autonomous agents that cooperate or compete under tasks designed to mimic the TV show "Squid Game." The sequence of tasks is arranged in the order of gradually increasing difficulty, from benign exchanges to dynamic, adversarial setups. The main contribution of the work is the standardization of roles, the protocol for dialogue (prompt templates), and the scoring/elimination metrics to assess model's reasoning and robustness performances. It also comes with a quantitative/qualitative study.

Experiments involve 50 open and proprietary LLMs evaluated across multiple rounds of interaction. Results show diverse model behaviors against scenarios and opponent types.

The claimed contribution is a general evaluation platform for interactive, multi-agent LLM behavior that, according to the work, extends beyond static benchmarks by probing “social” and “strategic” capabilities.

**Strengths:**

* The paper is reasonably well written, and contains no significant grammatical or format errors
* The use of illustration helps with the delivery and flow of the paper
* Individual mini benchmarks contain interesting operational constructions (e.g. Honeycombs) and are arguably salvageable

**Weaknesses:**

* The paper fundamentally constructs an elaborate, but highly artificial setup. The games are inspired by a fictional TV show, and has no natural correspondence to real-world LLM tasks.
* The claimed advantage of this evaluation, where "from what a model knows to what it can do under uncertainty," is not supported by evidence. The paper neither makes an attempt to clarify what aspects of existing evaluation protocols fails to satisfy the characteristics they care about, nor offer explanations on why their constructions solve the issue in a way that is externally verifiable (beyond that any audience of this venue would find them intuitively making sense, including myself, and possibly other reviewers).
* The experiment results offer no new insights. It's a well-understood and well-studied phenomenon that GPT-5 and Gemini 2.5 Pro are incredibly strong and generalizable models, so it is not surprising that they progress to "The Final." I would even imagine that if the results are independently run, it could correlate well with an **ensemble** of existing benchmarks (I am aware of Figure 8 and this is not the core of my objection and the authors need not actually run this to prove or disprove this point).

Benchmarks are very important to the venue and our community. It helps us understand where models excel and could improve on, especially these newer classes of foundation models that are huge and exhibit many kinds of emergent behaviors. However, benchmarks should also be designed to be accessible and insightful. The protocol should be self-contained, grounded in real capabilities and use cases we care about, and most important of all, falsifiable. The "Squid Game" analogy to me only appears as synthetic, invented game mechanics that are only valuable within the construct of the benchmark itself. As such, the surrounding justification in the work are mostly rhetorical rather than empirical, which would make the work unsuitable for publication.

**Questions:**

I don't have any questions as I understood the paper well due to it's great presentation. The issue with the manuscript is structural and I believe the paper is irrelevant to conference goals despite competent execution. I recommend strong reject.

---

### Official Review · Reviewer_73Yq · 2025-11-05

**Soundness:** 1
**Presentation:** 3
**Contribution:** 1
**Rating:** 2
**Confidence:** 5

**Summary:**

This paper introduces SQUID GAME, a benchmark for evaluating large language models in dynamic, adversarial, and resource-constrained settings, in contrast to traditional static and benign benchmarks.
Inspired by Netflix’s Squid Game, the benchmark comprises six elimination-based games designed to test diverse abilities.
Each round imposes resource constraints (e.g., token quotas) and information asymmetry, requiring models to plan and adapt dynamically.
Models are eliminated each round rather than scored absolutely, creating a battle-royale-style tournament.

**Strengths:**

1. Recasting LLM evaluation as an elimination-style, adversarial “game” offers a fresh, metaphor-driven approach to stress-testing model capabilities.

2. Experiments with 52 models provide rich comparative data, allowing broad insights into model behavior under dynamic constraints.

3. The paper is well-structured, figures are effective, and the motivation and empirical sections are easy to follow.

**Weaknesses:**

1. Although the paper introduces six interactive games, it provides insufficient formalization or reproducible implementation details, making it difficult for others to replicate or validate the experiments.

2. Correlation plots with existing benchmarks (e.g., LIVEBENCH, CHATBOT ARENA) remain anecdotal. The paper lacks formal statistical analysis (e.g., significance testing, regression, controlled ablations) and does not report error bars or variance beyond simple averages.

3. The paper provides limited discussion of prior work on interactive evaluation, multi-agent benchmarking, and adaptive test generation (e.g., DyVal, DARG, LLMArena). Without this context, the conceptual novelty of SQUID GAME is less clear.

DyVal: Dynamic Evaluation of Large Language Models for Reasoning Tasks

LLMArena: Assessing Capabilities of Large Language Models in Dynamic Multi-Agent Environments

DARG: Dynamic Evaluation of Large Language Models via Adaptive Reasoning Graph

**Questions:**

See Weaknesses

---

### Official Review · Reviewer_K9JW · 2025-11-05

**Soundness:** 3
**Presentation:** 3
**Contribution:** 3
**Rating:** 4
**Confidence:** 4

**Summary:**

Traditional benchmarks are failing to keep up with the rapid progress of large language models (LLMs) and may be compromised by data contamination, making it unclear whether models are truly reasoning or just recalling seen data. To address this, the paper introduces Squid Game, a dynamic, adversarial evaluation framework that tests LLMs in resource-limited, asymmetric, and interactive scenarios. It includes six elimination-style levels assessing diverse abilities like reasoning, planning, coding, and alignment. The study evaluates over 50 LLMs—one of the largest behavioral assessments to date—and finds generational performance shifts and evidence of shortcut strategies, suggesting contamination even in higher-level evaluations. Correlation analysis shows that dynamic testing complements static benchmarks, offering a more trustworthy way to assess LLM behavior.

**Strengths:**

- Innovative Evaluation Paradigm: Most existing LLM benchmarks are static and benign, while this work’s SQUID GAME pioneers an "elimination + dynamic adversarial" framework, filling the gap in dynamic adversarial LLM evaluation.
- Large-Scale Experiments: It evaluates 52 LLMs (28 proprietary like GPT-5/Gemini 2.5 Pro; 24 open-source like Qwen3/DeepSeek), the largest behavioral study in dynamic adversarial scenarios to date. It also identifies model exploitation of evaluation loopholes, providing empirical evidence for "methodological leakage" in static benchmarks.
- Evaluation Fairness: To avoid tokenizer biases, it uses "character count" for quota measurement and provides real-time resource feedback, simulating real-world constraints.
- Complementarity to Static Benchmarks: SQUID GAME shows low correlation (coefficient = 0.3675) with static benchmarks (e.g., CHATBOT ARENA, LIVEBENCH), effectively evaluating LLMs’ "dynamic capabilities" as a valuable supplement.

**Weaknesses:**

- Bias from Linear Elimination Order:SQUID GAME adopts a strict linear elimination sequence, where participants in subsequent levels are entirely determined by the survivors of the previous level. While this design simulates dynamic competition, it may introduce biases in evaluating the comprehensive capabilities of LLMs, specifically in two aspects:
  - Early levels focus on basic capabilities, which may eliminate models that perform poorly in basic tasks but excel in advanced capabilities. This results in the loss of opportunities to evaluate the comprehensive potential of such models, as the capabilities assessed across levels are not inherently sequential and are rather mutually independent to a certain extent.
  - The comprehensive capability of an LLM inherently relies on the synergy of multi-dimensional abilities. However, the fixed elimination order means that once a model fails in one capability dimension, it is denied the chance to compensate with strengths in other dimensions in subsequent levels.

- Insufficient Discriminability for High-Performing Proprietary Models: For top-performing proprietary models (e.g., Gemini 2.5 Pro and GPT-5) that can pass all levels of SQUID GAME, the current framework lacks effective metrics to distinguish their performance differences. This limits the framework’s ability to rank or compare the fine-grained capabilities of leading LLMs.

- Inapplicability of Competitive Mode to Certain Task Types: The competitive design of SQUID GAME may not be suitable for evaluating domain-specific capabilities such as mathematical performance. For instance, it remains unclear how to design levels within the competitive paradigm to effectively assess LLMs’ mathematical abilities, which are critical for many real-world applications.

- Subjectivity in Partial Level Evaluation and Dependence on External Data:
Subjectivity Risks: The "Tug of War" level employs a hybrid judging system to determine shifts in debate positions. Although the work mentions that the agreement rate between LLM judges and humans exceeds 95%, human experts’ subjective judgments may still be influenced by factors such as debate style and expression bias. No detailed calibration methods for human judgment are provided, raising potential concerns about evaluation objectivity.

- External Data Dependence: Key tasks in some levels rely on external datasets: the long tasks in the "Red-Green Light" level are sourced from LIVEBENCH, and the code for the "Sugar Honeycombs" level is derived from HUMANEVAL and LIVECODEBENCH. If these external datasets suffer from data contamination, the evaluation results may be biased. However, the work does not elaborate on decontamination processes for the external data used.

**Questions:**

Listed in Weakness.

---

### Official Review · Reviewer_QQY9 · 2025-11-06

**Soundness:** 2
**Presentation:** 3
**Contribution:** 3
**Rating:** 6
**Confidence:** 2

**Summary:**

The paper presents Squid Game, a novel benchmark for evaluating LLMs under progressively challenging conditions. The benchmark is inspired by the Squid Game tournament format: it consists of six sequential elimination-style tasks. These tasks range from benign instruction-following scenarios to highly adversarial, zero-sum contests, including safety alignment tests, thus covering a broad spectrum of LLM robustness. Unlike traditional LLM benchmarks, Squid Game places models in interactive “gameplay” against other LLMs, with resource constraints (e.g., limited tokens or API calls) and information asymmetry (e.g., models may have incomplete information or see others’ outcomes) to mimic realistic pressure situations. The authors evaluate over 50 LLMs on this benchmark.

**Strengths:**

- Squid Game is an interesting idea with strong motivations. For example, many static benchmarks are nearing saturation with top models, and data contamination can inflate scores. Squid Game directly tackles this by introducing information asymmetry and dynamic adversarial evaluation. The benchmark shifts the question from “What does the model know?” to “How does the model act under uncertainty and competition?”. This perspective is fresh.
- Squid Game’s six levels are carefully chosen to exercise a wide range of abilities, from basic instruction following and code generation to complex reasoning, strategy, and safety alignment. By integrating these diverse tasks into one suite, the benchmark provides a holistic evaluation of LLM behavior under different conditions. This breadth is valuable, as it can reveal trade-offs that single-domain benchmarks would miss.
- The authors conduct an extensive study, testing 52 LLMs (28 proprietary, 24 open-source) across all scenarios. The experimental methodology appears sound: the paper reports repeating the tournament multiple times to get stable pass rates, and it analyzes results from multiple angles (performance by level, elimination rounds, etc.).

**Weaknesses:**

- The Squid Game benchmark introduces a fairly complex evaluation setup with multiple games and roles. The paper would benefit from clearer descriptions of each game’s rules and scoring. For example, how exactly is a “win” determined in the Tug-of-War debate, or what constitutes success in the Marbles game? A more explicit explanation of the evaluation criteria for each level would improve clarity.
- By design, this benchmark is resource-heavy. Running head-to-head evaluations on 50+ models with multiple rounds is a non-trivial undertaking.  In the meantime, it is difficult for others to verify results or try Squid Game on their own models. Even with code available, reproducing the full tournament may require significant compute and cost, especially for API models. The paper could be strengthened by discussing ways to partially evaluate models or reuse results to lower the barrier (e.g., using a fixed set of reference opponents or fewer rounds for a quick assessment).
- Some of the Squid Game tasks involve qualitative or complex outcomes that are harder to auto-grade than standard QA accuracy. For example, determining the winner in a debate or a safety-red teaming exchange may involve subjective judgments or require a trusted referee model/human. If the current evaluation relied on humans or ad-hoc heuristics at any stage, that could introduce evaluation bias. The paper would benefit from clarifying the evaluation protocol for subjective tasks (e.g., was GPT used to judge debates, or were there predefined metrics?).
- The elimination-based ranking is relative, which means a model’s performance is measured against others in the pool. This raises a fundamental question: how will the benchmark handle the evaluation of a single new model in the future? The paper suggests running new seasons, but if a user wants to test one model’s Squid Game performance, do they need to run a full tournament with a large cast of models? The authors should address how stable the evaluation is and provide more information on variance would help. If the outcome for a given model can vary depending on bracket pairings or luck, especially in stochastic tasks like Red-Green Light with random interruptions, that could be a weakness in terms of reliable benchmarking.

**Questions:**

Please refer to the weaknesses.

---

### Official Review · Reviewer_YrXw · 2025-11-07

**Soundness:** 2
**Presentation:** 1
**Contribution:** 3
**Rating:** 2
**Confidence:** 4

**Summary:**

This paper introduces SQUID GAME, a dynamic and adversarial benchmark for evaluating Large Language Models (LLMs). It frames the evaluation as an elimination-style tournament inspired by the Netflix series, consisting of six competitive, resource-constrained games designed to test abilities like instruction-following, reasoning, and safety alignment.

**Strengths:**

- The core idea of using a dynamic, adversarial tournament for LLM evaluation is highly novel, creative, and engaging.

- The paper conveys a sense of enthusiasm for the project, suggesting a highly motivated research effort.

**Weaknesses:**

* **Missing Summative Ranking:** The paper's "Squid Game" theme creates a strong expectation for a final tournament outcome. A significant weakness is the lack of a clear figure or table showing the **final ranking** of all evaluated models, or a leaderboard aggregated over the 20 independent runs. Presenting a clear winner and runner-ups would be a crucial and satisfying addition to complete the paper's narrative.
* **Insufficient Qualitative Analysis:** The paper is currently focused on quantitative results (survival rates) but lacks discussion on *why* models behaved as they did. The analysis would be substantially strengthened by including **qualitative examples** of model failures and successes, illustrating their reasoning processes, particularly in complex games like 'Marbles' or 'Tug of War'.
* **Clarity and Polish of Prose:** The paper would benefit from a thorough round of **copy-editing** to improve clarity and professionalism. Some sentences are difficult to parse and detract from the core message (e.g., the paragraph beginning: "Games provide an intuitive and comparable signal of success...").
* **Presentation and Use of Space:** The paper's layout and use of space could be improved. **Figure 6**, in particular, is very large and dense, making it difficult to interpret in the main paper; its space might be better utilized for the qualitative discussion mentioned above, or the figure could be moved to the appendix.
* **Motivation for Some Tasks:** While most games have a clear purpose, the motivation for tasks like the **'Glass Stepping Stones'** could be better justified in terms of the specific LLM capabilities it tests.
* **Occasional Overstatement:** Claims in the text should be carefully rephrased to avoid overstatement. For instance, citing a single paper to suggest the topic "has been widely studied" or describing the 'Glass Stepping Stones' game as a "rich simulation" requires stronger evidence or more modest language.

**Questions:**

* Given the highly competitive, elimination-style nature of the evaluation, was the **meta-game context** (i.e., being in an elimination tournament with other models) explicitly communicated to the LLMs in their initial prompts?
* The original "Squid Game" featured tasks with elements of unfairness. Have the authors considered **introducing a subtly unfair task** to test whether LLMs can identify, adapt to, or 'complain' about the unfair mechanics? This could offer a fascinating test of a model's meta-awareness.
* In the analysis of the 'Red-Green Light' game, you observe that **lightweight models** (e.g., MINI, FLASH versions) perform better than their standard versions. Could you please provide a technical explanation for this finding? Why would a smaller model have an intrinsic advantage in this instruction-following and interruption-handling task?

---

### Official Review · Reviewer_YF3o · 2025-11-07

**Soundness:** 2
**Presentation:** 2
**Contribution:** 1
**Rating:** 2
**Confidence:** 4

**Summary:**

This paper introduces a dynamic LLM benchmarking framework inspired by the Netflix show "Squid Game." It involves multiple rounds of elimnation between different language models based on different tasks with the goal of identifying stronger and weaker models. The authors apply this framework to 52 LLMs. A comparison to existing general evaluation frameworks such as ChatBot Arena shows only moderate correlation.

**Strengths:**

* Devising novel dynamic evaluation methods is an important area, given that training data contamination and inaccessibility of the training data of many current models makes it difficult to assess which abilities result from memorization and which ones result from generalization abilities.
* The paper evaluates a wide range of LLMs.

**Weaknesses:**

* My general impression is that the authors focused most on their energy to come up with tasks that could be applied to LLMs and that mimic tasks on the Netflix show "Squid Game". While it can be beneficial to take inspiration from very different domains, the current work fails to motivate why these rounds are reasonable.
* It is unclear what this benchmark is supposed to highlight. Given the multitude of different LLMs, many of which have been optimized for different aspects, it seems misguided trying to find THE best LLM. It is also not clear what exactly the authors would consider the best LLM.
* In general, benchmarks are designed so that one can (barring issues such as memorization) draw conclusions about certain model abilities, e.g., on coding tasks, common sense reasoning, or logical reasoning. I was not really able to draw any such conclusions from the results presented in this paper (and similarly, the conclusions that the authors draw on pages 7 and 8, are also very vague and do not seem helpful to model developers or users who want to decide which model to use).
* When developing a benchmark, one should illustrate that it measures something useful. Now if this benchmark had had a very strong correlation with human benchmarks like Charbot Arena, it would have been possible to argue that the presented benchmark can be run with less human intervention. However, the low correlation is explained with the present benchmark and Chatbot Arena targeting "orthogonal dimensions" without explaining what these dimensions are. This makes me think that the present benchmark may not measure anything useful.
* The relation to some other dynamic benchmarks like "DynaBench" (https://dynabench.org/) is not explained in the paper.

**Questions:**

* What is this benchmark supposed to measure exactly?
* What can we take away from a model winning or losing a specific round?

---

### Note · Authors · 2026-01-02

I have read and agree with the venue's withdrawal policy on behalf of myself and my co-authors.